# Globally correlated conformational entropy underlies positive and negative cooperativity in a kinase's enzymatic cycle

Yingjie Wang[1,2], Manu V.S.[2], Jonggul Kim [1,2,4], Geoffrey Li[1,5], Lalima G. Ahuja[3], Philip Aoto [3], Susan S. Taylor [3] & Gianluigi Veglia[1,2]

Enzymes accelerate the rate of chemical transformations by reducing the activation barriers of uncatalyzed reactions. For signaling enzymes, substrate recognition, binding, and product release are often rate-determining steps in which enthalpy-entropy compensation plays a crucial role. While the nature of enthalpic interactions can be inferred from structural data, the molecular origin and role of entropy in enzyme catalysis remains poorly understood. Using thermocalorimetry, NMR, and MD simulations, we studied the conformational land-scape of the catalytic subunit of cAMP-dependent protein kinase A, a ubiquitous phosphoryl transferase involved in a myriad of cellular processes. Along the enzymatic cycle, the kinase exhibits positive and negative cooperativity for substrate and nucleotide binding and product release. We found that globally coordinated changes of conformational entropy activated by ligand binding, together with synchronous and asynchronous breathing motions of the enzyme, underlie allosteric cooperativity along the kinase's cycle.

[1] Department of Chemistry, University of Minnesota, Minneapolis, MN 55455, USA. [2] Biochemistry, Molecular Biology, and Biophysics, University of Minnesota, Minneapolis, MN 55455, USA. [3] Department of Chemistry and Biochemistry, and Pharmacology University of California at San Diego, La Jolla, CA 92093, USA. [4] Present address: Department of Biophysics and Howard Hughes Medical Institute, University of Texas Southwestern Medical Center, Dallas, TX 75390, USA. [5] Present address: Department of Biochemistry, Vanderbilt University School of Medicine, Nashville, TN 37240, USA. These authors contributed equally: Yingjie Wang, Manu V.S., Jonggul Kim. Correspondence and requests for materials should be addressed to G.V. (email: vegli001@umn.edu)

The fine balance of enthalpy and entropy dictates the free energy of substrate binding and product release in enzymatic catalysis. How these two contributions drive enzymatic catalysis remains unclear. In the past decades, X-ray crystallography has greatly contributed to our understanding of how an enzyme works, providing an enthalpic view about the origins of the interactions that govern the catalytic cycle. Although the presence of conformational dynamics in enzymes can be inferred from the resolution of the electron density maps, X-ray data fall short to provide any quantitative information on the time scale of motions and their link to catalysis. In contrast, nuclear magnetic resonance (NMR) spectroscopy is the experimental method of choice to monitor molecular fluctuations at the atomic level[1,2]. Seminal work by different groups has revealed the involvement of specific modes of motions in enzymatic activity[2–6]. While NMR-derived nanosecond–millisecond motions are likely not to be involved in the chemical step of catalysis[7], there is strong evidence that ligand binding affinities and kinetics of structural transitions are directly modulated by dynamics in the picosecond-to-nanosecond and micro-to-millisecond time scale, respectively[8–12]. Nonetheless, it remains unclear whether structural fluctuations during enzymatic catalysis are randomly distributed or are concerted to maximize catalytic efficiency.

Here, we analyze the conformational energy landscape of the catalytic subunit of cAMP-dependent protein kinase A (PKA-C) along its reaction coordinates using isothermal titration calorimetry (ITC) and NMR spectroscopy. The PKA-C architecture is highly conserved (Fig. 1a), making it a benchmark for studying the mechanisms of signaling and regulation for the entire AGC kinase family[13]. PKA-C is a signaling enzyme that controls vital cellular processes such as skeletal and cardiac muscle contractility, cell proliferation, and memory[14]. During the enzymatic cycle, PKA-C adopts several conformational states corresponding to different ligand-bound forms: apo, ATP-bound, ATP and substrate bound, ADP and phospho-product bound, and ADP-bound (Fig. 1a, b and Supplementary Fig. 1)[15,16]. The overall turnover rate of the kinase is approximately $20\,s^{-1}$, with a fast phosphoryl transfer (chemical step, ~$500\,s^{-1}$) and a rate-determining ADP release step[17]. PKA-C binds nucleotide and unphosphorylated substrate via positive K-type cooperativity, while the phosphorylated substrate and ADP display a negative binding cooperativity, conceivably to favor phospho-product release. Our group and others suggested that conformational dynamics of PKA-C may drive the catalytic cycle[18–20]. Using nuclear magnetic spin relaxation measurements of the methyl-bearing side chains, we examined the dynamic response of the kinase to ligand binding. We found that highly coordinated subnanosecond dynamics underlie both positive and negative binding cooperativity, revealing that changes in conformational entropy fine-tune ligand binding affinity throughout the enzymatic cycle. Using methyl-TROSY relaxation dispersion (RD) measurements, we discovered that synchronous breathing motions of the enzyme in the micro-to-millisecond time scale underscore positive binding cooperativity between ATP and substrate; while asynchronous dynamics characterize negative cooperativity between ADP and phosphorylated product. Changes in conformational entropy are globally distributed throughout the enzyme and not limited to active site between the two lobes. These observations were further corroborated using extended molecular dynamics simulations (>5 μs) on the PKA-C/ATP/substrate complex and the PKA-C/ADP/phospho-product. Taken together, our findings reveal that globally correlated motions along the kinase enzymatic cycle drive allosteric cooperativity and efficient turnover.

## Results

**Agreement of calorimetric and NMR-derived PKA-C energy landscape.** Prior to phosphoryl transfer (chemical step), the kinase must bind both ATP and substrate. This ternary form represents a catalytically committed state, primed for the phosphotransfer step. Kinetically, there is little discrimination between the binding of the nucleotide or the substrate peptide (Kemptide) to the apo enzyme[17]. These two binding events exhibit positive K-type cooperativity, wherein binding of the nucleotide or substrate increases the affinity of the enzyme for the second ligand[21]. While thermodynamics allows for either the nucleotide or substrate to bind the apo kinase, given the high-ATP concentration in the cell, it is likely that the nucleotide binds the enzyme first and is followed by the substrate[17]. The thermodynamics of ATP and substrate binding to PKA-C has been extensively studied and it has been shown that nucleotide and the pseudo-substrate protein kinase inhibitor (PKI) peptide display approximately 400-fold binding cooperativity[21]. For our studies, we utilized a peptide derived from PKI as a substrate, where the alanine in the P position is substituted by a serine ($PKS_{5-24}$, Supplementary Fig. 2). The $PKS_{5-24}$ is an ideal mimic for the natural substrate, and given its high-binding affinity, it locks the enzyme in a committed state. Recent X-ray structures have shown how $PKS_{5-24}$ can be trapped with nucleotide in the binding groove of PKA-C, thereby allowing one to follow the phosphoryl transfer reaction[22,23].

To map the changes in free energy of binding ($\Delta\Delta G$) upon ligand binding, we carried out ITC experiments (Supplementary Figs. 3 and 4). Using the apo form as a reference state, we found that the binding of ATPγN lowers the free energy by approximately $6\,kcal\,mol^{-1}$ [24]. A more significant change is observed upon binding both ATPγN and $PKS_{5-24}$ ($\Delta\Delta G \sim -15\,kcal\,mol^{-1}$). While ADP binding causes a change of approximately $-9\,kcal\,mol^{-1}$, the binding of ADP and phosphorylated PKS ($pPKS_{5-24}$) differs by only $1\,kcal\,mol^{-1}$. We utilized the ITC data to quantify the cooperativity coefficient ($\sigma$) (Table 1). Interestingly, $PKS_{5-24}$ exhibited a higher affinity for the apo PKA-C compared to $PKI_{5-24}$ ($K_d = 0.8\,\mu M$), which could be explained by a more favorable enthalpic contribution due the presence of the hydroxyl acceptor in $PKS_{5-24}$. As found for other substrates, ATPγN or ATPγC saturated kinase binds $PKS_{5-24}$ with higher affinity than the apo form with $\sigma$ of 2.8 and 4.4, respectively. The latter is consistent with values found in literature for other substrates[18,20]. We then estimated the cooperativity between ADP and $pPKS_{5-24}$, which bind the apo PKA-C with a $K_d$ of 10 and 3.5 μM, respectively. Unlike ATP and substrate, ADP and $pPKS_{5-24}$ show negative binding cooperativity. In fact, the PKA-C/ADP complex binds $pPKS_{5-24}$ with a $K_d$ of 7.4 μM. Although the negative cooperativity is only twofold, the trend for the binding cooperativity follows the same direction of the standard substrate Kemptide[25]. The change of the sign for the K-type allosteric cooperativity upon substrate phosphorylation has been reported in other kinases[26], suggesting that the γ-phosphate plays a critical role for both binding affinity and allosteric cooperativity.

To analyze the structural changes of PKA-C under different ligated states, we mapped the linear response of PKA-C methyl chemical shifts upon binding ATPγN, ADP, $PKS_{5-24}$, and $pPKS_{5-24}$ using CONCISE[27] (Fig. 1c, Supplementary Figs. 5–7). The linearity of the chemical shifts suggests that the kinase is in a fast conformational equilibrium between open and closed states, represented by the apo and $PKI_{5-24}$ bound forms, respectively[28]. Upon addition of ATPγN, the conformation of the kinase shifts toward the intermediate state. The ternary complexes with nucleotide and $PKS_{5-24}$ both in the phosphorylated and unphosphorylated forms further shift the probability distribution of the

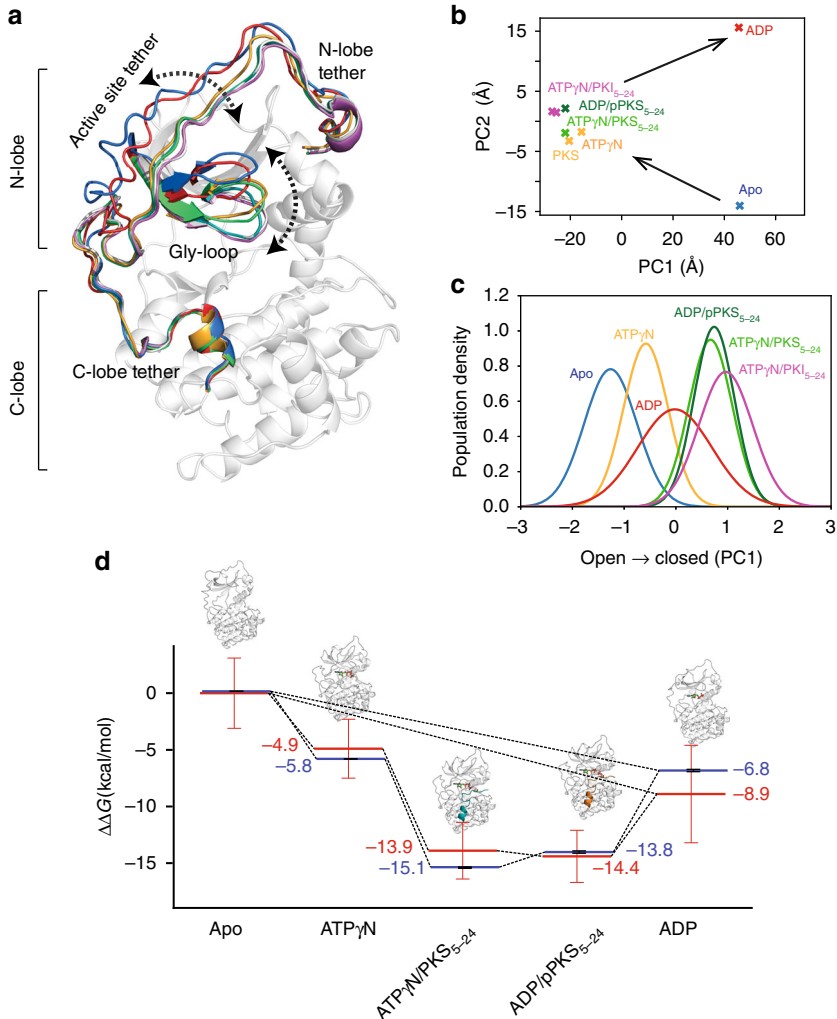

**Fig. 1** Conformational transition of PKA-C during turnover. **a** Superposition of the X-ray crystal structures of PKA-C in the apo (PDB code: 4NTS), binary complex (ATPγN-bound, PDB code: 1BKX), ternary complex (ATPγN and PKS$_{5-24}$, PDB code: 4DG0), ternary/exit complex (ADP and pPKS bound, PDB code: 4IAF), and binary (ADP-bound, PDB code: 4NTT). Dotted arrows indicate the major domains involved in large amplitude motions determining opening and closing of the nucleotide site and substrate hub. **b** Principal component analysis (PCA) with the two main components indicating the structural transitions in the crystal structures of PKA-C for different ligated states, where PC1 and PC2 involve distinct collective motions throughout the protein (illustrated in Supplementary Fig. 1). **c** CONCISE plot showing the probability distribution curves of the methyl chemical shifts for the different states along the open-to-close trajectories. **d** Changes in the free energy of binding (ΔΔG) determined by ITC (blue) measurements (mean ± SD, $n = 3$) compared with the ΔΔG obtained from the CONCISE analysis (red) that estimates the relative population of the bound states. The CONCISE data are obtained from probability density distributions of the NMR chemical shifts for the ILV methyl groups of PKA-C, where the error bar reports the 90% confidence interval. The data of the CONCISE plots are reported in the Source Data file

methyl resonances toward the closed state. The slightly narrower probability distribution obtained in response of PKS$_{5-24}$ binding indicates that the transition toward the Michaelis complex occurs in a more coordinated fashion than the corresponding complex with PKI$_{5-24}$. This is probably due to more favorable interactions between the phosphate group of ATPγN and the phosphoryl acceptor of the Ser in the substrate within the active site (Fig. 1c). The superposition of the probability distributions for both ternary complexes with ATPγN/PKS$_{5-24}$ and ADP/pPKS$_{5-24}$ indicates that the average conformations of these complexes are similar. Accordingly, the crystal structures of PKA-C with ATPγN/PKS$_{5-24}$ and ADP/pPKS$_{5-24}$ show distances between S53 in the Gly-rich loop and G186 in the DFG motif of 11.4 and 10.4 Å, respectively, which define the kinase's crystallographic closed state[29,30]. To calculate the free energy of binding, we converted the probability distributions into populations[27]. Remarkably, the values of free energy of binding obtained from the NMR

spectroscopy and chemical shift changes of ΔΔG agree quantitatively with those obtained from thermocalorimetry (Fig. 1d, Supplementary Fig. 4). Only small differences are observable between the ternary complex with ATPγN and PKS$_{5-24}$ and the binary complex with ADP. Yet, these values are within the standard deviations obtained from the probability density of the methyl chemical shifts. Collectively, these data indicate that the energy landscape sampled via NMR spectroscopy has a direct correspondence with the global free energy deduced from the thermodynamic measurements.

To map the site-specific allosteric changes along the reaction coordinates, we used the chemical shift covariance analysis (CHESCA), which traces the allosteric network upon ligand binding[31,32]. CHESCA correlation matrices show a remarkably high degree of covariance between residues located in the two lobes upon ligand binding (Supplementary Figs 8, 9). These data signify that both nucleotide and substrate binding bring

**Table 1 The affinity, enthalpy, entropy and free energy of substrate (PKS$_{5-24}$), phosphorylated substrate (pPKS$_{5-24}$), and nucleotide (ADP) binding to PKA-C without (Apo) or with nucleotide (mean ± SD, $n = 3$)**

| | $K_D$ (μM) | $\Delta G$ (kcal mol$^{-1}$) | $\Delta H$ (kcal mol$^{-1}$) | $T\Delta S$ (kcal mol$^{-1}$) |
|---|---|---|---|---|
| **+PKS$_{5-24}$** | | | | |
| Apo | 0.80 ± 0.03 | −8.38 ± 0.03 | −19.10 ± 0.26 | −10.72 ± 0.26 |
| ATPγC | 0.29 ± 0.01 | −8.99 ± 0.03 | −13.56 ± 0.22 | −4.60 ± 0.24 |
| ATPγN | 0.18 ± 0.00 | −9.27 ± 0.01 | −20.75 ± 0.39 | −11.48 ± 0.39 |
| **+pPKS$_{5-24}$** | | | | |
| Apo | 3.52 ± 0.47 | −7.50 ± 0.08 | −12.40 ± 0.65 | −4.90 ± 0.73 |
| ADP | 7.43 ± 0.67 | −7.05 ± 0.05 | −10.92 ± 0.03 | −3.87 ± 0.06 |
| **+ADP** | | | | |
| Apo | 10.58 ± 1.81 | −6.85 ± 0.10 | −5.32 ± 0.20 | 1.53 ± 0.10 |

together important catalytic motifs and organize the active site for catalysis in a cooperative manner. Importantly, several residues located in the C-spine (L172 and L227), the hydrophobic core (V98, L103, I150, and I180) undergo correlated changes upon ATPγN binding to the enzyme[33]. These chemical shifts further emphasize the concerted structural changes hypothesized based on community analysis from molecular dynamics simulations[34,35]. Overall, the methyl chemical shift changes revealed a global response of the kinase upon ligand binding not only for the amide backbone group[28], but also for the methyl-bearing side chains, showing highly cooperative structural changes nearby the binding site and in remote locations of the enzyme.

**Conformational entropy underlies binding cooperativity.** Recent groundbreaking papers have established a direct correlation between protein's conformational entropy and binding affinity[10,11,27,36]. Specifically, both experimental[37] and computational[38] works support the original hypothesis that allosteric cooperativity can be modulated by changes in conformational motion alone[39]. Therefore, we sought to investigate the role of conformational entropy in the positive and negative binding cooperativity in the kinase's enzymatic cycle. ITC data show a significant difference in the total entropy change between ATP/substrate and ADP/phospho-substrate binding to PKA-C. To determine the conformational entropy, we measured the subnanosecond dynamics of the methyl groups[10,36,37,40] using relaxation violated coherence transfer cross-correlation experiments[41,42]. First, we estimated the isotropic rotational correlational time ($\tau_c$) of the protein using dynamic light scattering (DLS) (Supplementary Fig. 10). For the kinase complexes with ADP, ADP/pPKS$_{5-24}$ and ATPγN/PKS$_{5-24}$, we obtained $\tau_c$ values of 29.6, 25.3, and 25.3 ns, respectively, which are consistent with previously published data[20,43]. Then, we used these $\tau_c$ values to determine methyl group order parameters ($O^2$) for the ILV methyl-bearing side chains[41,42]. Fig. 2 shows the $O^2$ values for the methyl-bearing side chains mapped onto the various crystal structures along the enzymatic cycle with the respective free energy, entropy, and enthalpy of binding as well as the conformational entropy. The residue-specific changes of $O^2$ between the various ligated forms can be fully appreciated from Supplementary Fig. 11. The *apo* enzyme corresponds to the most dynamic state and undergoes an overall rigidification upon nucleotide binding (average $\Delta O^2 > 0$). In this step, the enzymes prepays the entropic cost for the subsequent binding of the substrate, a phenomenon common to other protein–protein interactions[9,38]. Remarkably, the most significant changes involve methyl groups buried in the hydrophobic core spanning both the C- and R-spines[33]. Upon binding PKS$_{5-24}$, the $\Delta O^2$ for most of

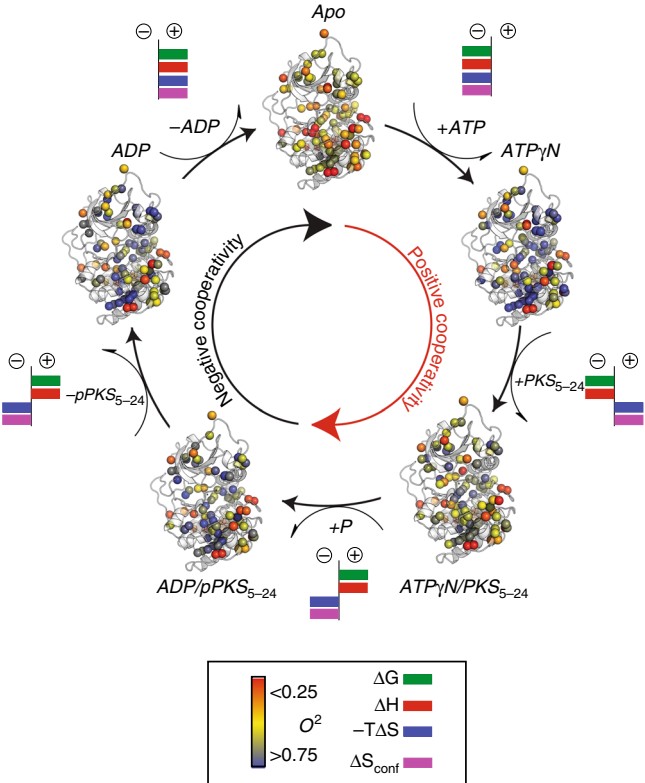

**Fig. 2** Changes of conformational entropy of PKA-C along the coordinate of reactions. Ribbon representation of the different ligated conformations of PKA-C along the coordinate of reactions as obtained from X-ray crystallography: apo (PDB code: 4NTS), binary complex (ATPγN-bound, PDB code: 1BKX), ternary complex (ATPγN and PKS$_{5-24}$ bound, PDB code: 4DG0), ternary/exit complex (ADP and pPKS$_{5-24}$ bound, PDB code: 4IAF), and binary (ADP-bound, PDB code: 4NTT). The methyl group $O^2$ obtained for the different forms are indicated with a colored gradient from the most rigid (blue) to the most mobile (red). The ligands are omitted from the figure for clarity. For each transition, we indicated the signs for total free energy (green), enthalpy (red), and entropic penalty (blue) as obtained from ITC measurements as well as the average sign for conformational entropy (purple) reflected by $\Delta O^2$ from NMR measurements

the methyl groups becomes negative, indicating that substrate binding increases the conformational dynamics throughout the entire enzyme. A similar behavior was observed for the amide backbone resonances of PKA-C upon binding a phospholamban peptide[28]. For the PKA-C/ADP/pPKS$_{5-24}$ complex, which mimics the product formation, there are only a few methyl groups that become more rigid in the subnanosecond time scale, specifically, I46 at the beginning of the β$_1$ strand and L74 in the β$_3$ that is located nearby the salt bridge formed by K72 and E91, a signature of active kinases[44]. Upon binding the substrate, there is a global increase of motions ($\Delta S > 0$, average $\Delta O^2 \sim -0.05$). In contrast, upon binding pPKS$_{5-24}$ the increase in motion is substantially lower than for the unphosphorylated substrate. The most significant differences in dynamics are located in the side chains that line the nucleotide and substrate binding sites, e.g., L74, I209, and I250 (Supplementary Fig. 11). Remarkably, these changes in the order parameters throughout the enzymatic cycle are globally correlated as shown by the covariation of the methyl order parameters[45]. The two-dimensional map of the covariance is reported in Fig. 3a and mapped on the structure in Fig. 3b. Indeed, the changes of the methyl groups throughout the entire cycle are highly correlated and extend far away from the nucleotide and substrate binding sites. This high level of

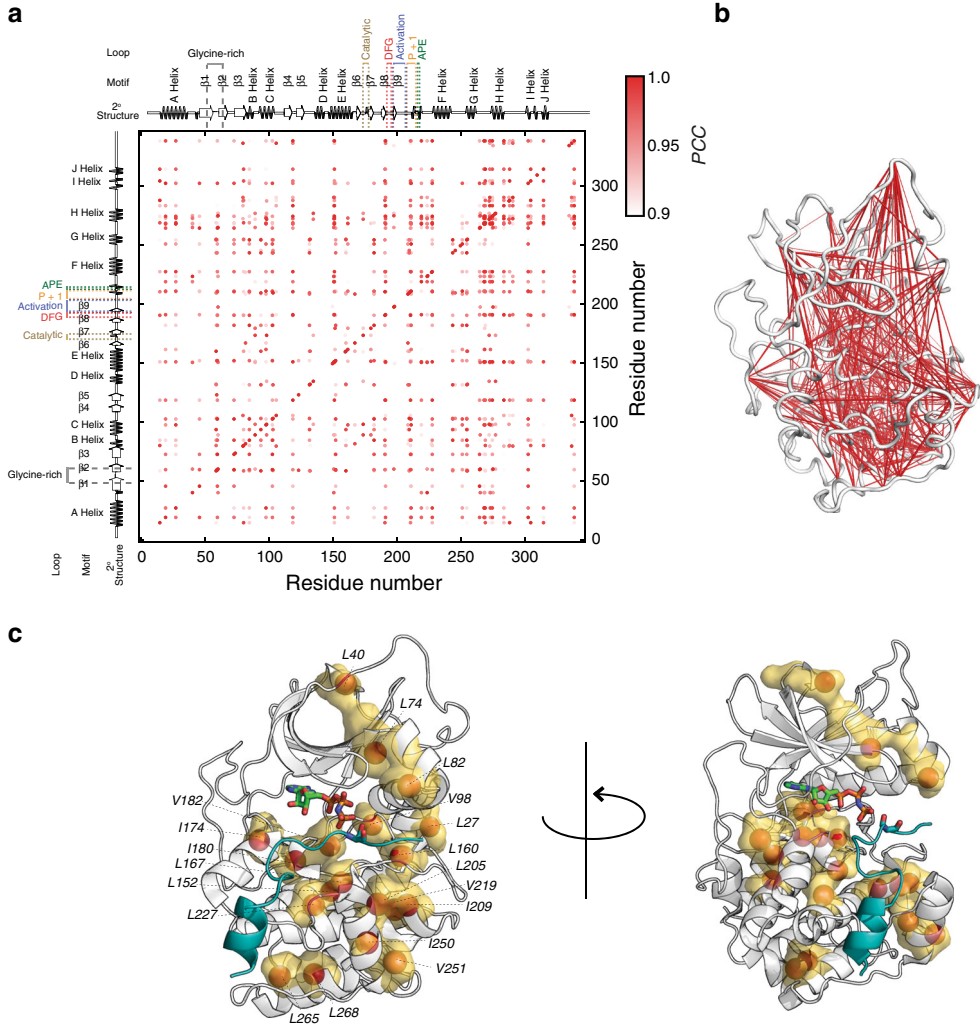

**Fig. 3** Correlated motions orchestrate positive and negative cooperativity. **a** Covariance plot indicating the coordinated changes of methyl group order parameters ($O^2$) throughout the entire kinase. The diagonal peaks in the plots represent the assigned methyl groups, while cross peaks indicate the degree of correlation of the methyl group $O^2$ using Pearson coefficient as a metric. Methyl group dynamics changes are considered coordinated when the Pearson coefficient is greater than 0.9. **b** Network plot connecting the methyl groups the highest values of covariance. **c** Mapping of the methyl groups that show a significant difference in order parameters upon forming the ternary complexes, i.e., PKA-C/ATPγN/PKS$_{5-24}$ and PKA-C/ADP/pPKS$_{5-24}$, from the binary complexes bound with nucleotide, respectively (see Supplementary Fig. 12)

covariation for $\Delta O^2$ indicates that nucleotide binding affects the global conformational entropy of the kinase and thereby the binding affinity of the kinase for the substrate. A reverse, correlated trend of $\Delta O^2$ is observed for the second part of the cycle, which features the release of phosphorylated product and ADP. The difference $\Delta\Delta O^2$ upon formation of the Michaelis Menton and product formation complex is largely positive (Supplementary Fig. 12), signifying reduction in entropy and binding affinity, underscoring negative cooperativity. By mapping the most affected methyl groups on the kinase structure, it is possible to note that the changes in order parameters are distributed in both lobes and across the hydrophobic spines, lining the face of the kinase in direct contact with nucleotide and substrate (Fig. 3c).

Molecular dynamics simulations were used to calculate the conformational entropy of the residues and corroborate the NMR data (Fig. 4). As for the NMR results, the changes of conformational entropy are redistributed throughout the enzyme; rather than being localized around the active site. The high overall entropy of the apo state of PKA-C is lowered by ~30 cal mol$^{-1}$ K$^{-1}$ in the enzyme bound to ATP (Supplementary Table 2); meanwhile, there is only an increase of entropy of ~ 7 cal mol$^{-1}$ K$^{-1}$ for the kinase going

from the committed state to the product state. Both values yield quantitative agreement between MD and NMR estimations (Supplementary Table 2). Furthermore, the ternary form following the phosphotransfer process shows an increase of conformational entropy of the phospho-product (Fig. 4). Collectively, our results support the theory that conformational entropy is intimately related to ligand binding affinity, and more importantly, plays a central role in orchestrating positive and negative binding cooperativity.

**Synchronous and asynchronous μs–ms motions during turnover.** To understand the involvement of the slow conformational dynamics (μs–ms timescale) in the various steps of the PKA-C catalytic cycle, we measured the RD of the methyl groups of the kinase using single-quantum CPMG experiments. We fit the dispersion curves using a two-state exchange model and extracted thermodynamics ($p_A$ and $p_B$), kinetics ($k_{ex}$), and structural ($\Delta\omega$) information (Supplementary Tables 4–6)[46–48]. To cluster the methyl groups based on the kinetics of exchange, we introduced dynamic correlation (DyCorr) maps that depict how methyl groups are dynamically correlated within the kinase. This is an

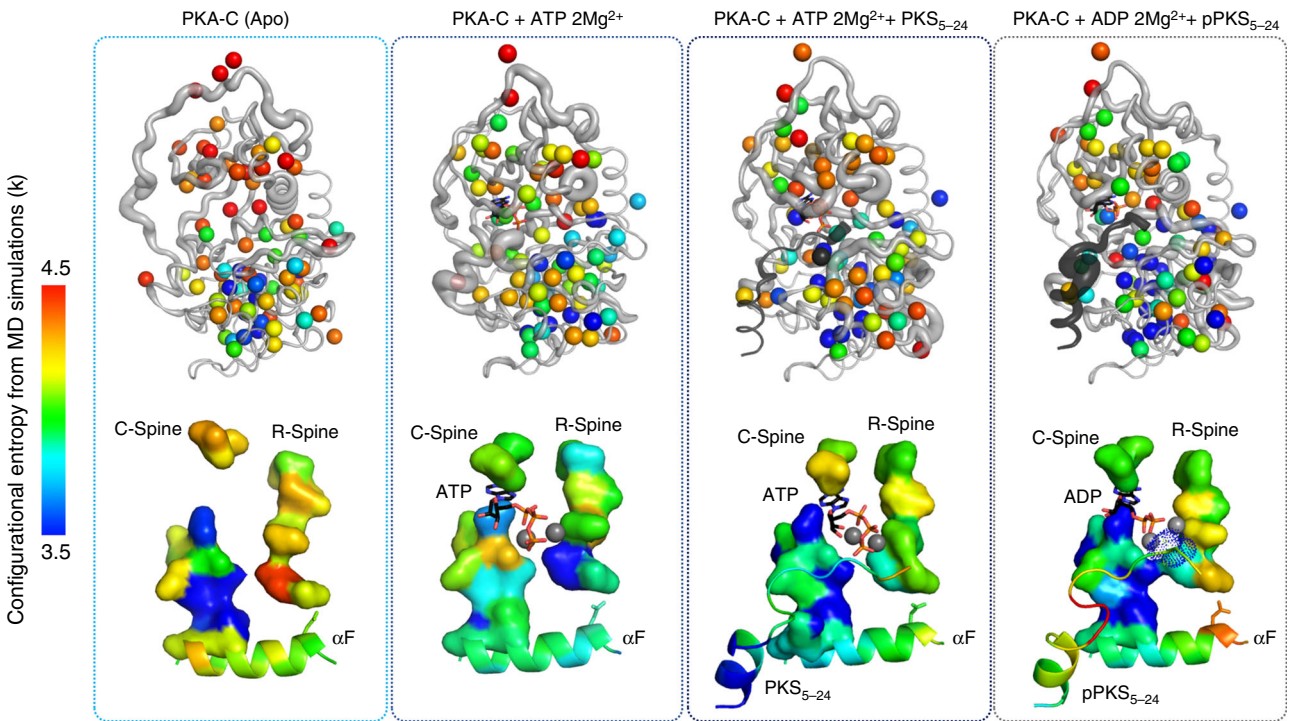

**Fig. 4** Conformational entropy analysis of PKA-C based on MD simulations. Conformational entropy analysis of the main chain and side chains of the Apo, ATP, ATP + substrate and the ADP + phospho-product forms of PKA-C. The altered entropy of the enzyme upon binding ATP primes the enzyme for substrate binding, whereas the change of entropy in the ADP + phospho-product state primes the enzyme for product release. Changes in entropy are distributed globally through the enzyme

alternative approach to the common clustering methods, which group the resonances based on exchange rates and populations[49]. To obtain the DyCorr maps, we fit each dispersion curve independently using the Carver-Richards equation, calculated $k_{on}$ and $k_{off}$ for each group and clustered residues based on the minimal distances in the $k_{on}/k_{off}$ phase space (see Methods). Figure 5a shows the methyl groups for which dispersion curves were measured. The color code indicates the extent of $R_{ex}$ measured by the CPMG experiments, which is mapped onto the corresponding X-ray crystal structures for each kinase state. In the adjacent Fig. 5b, the DyCorr maps are arranged according to the enzymatic cycle and indicate the covariance of the chemical exchange in the µs–ms timescale across the kinase structure. In the *apo* state, the kinase shows several methyl groups undergoing conformational exchanges that are interspersed throughout both the small and large lobes. The corresponding DyCorr map shows that several methyl groups undergo motions with similar exchange constants. Highly correlated motions (i.e., DyCorr coefficients greater than 0.9, orange-red) are present in the $\beta_2$–$\beta_3$ connecting loops, the αB–αC loop as well as the region connecting the αC with $\beta_4$ (Supplementary Figs. 13 and 14). These methyl groups fluctuate on the same time scale as the methyl groups of the catalytic and activation loops. Upon binding the nucleotide, there is a general reorganization of the conformational dynamics, with synchronous motions encompassing residues in the nucleotide binding pocket and the substrate binding groove (Supplementary Fig. 15). Structural elements peripheral to the binding pockets such as the C-terminal tail also move synchronously. In this state, the enzyme adopts a *dynamically committed* state primed for substrate binding. Unlike the Michaelis complex obtained with a phospholamban derived peptide[20,28], the PKA-C/ATPγN/PKS$_{5-24}$ complex does not show conformational exchange in the time regime sensitive to CPMG experiments. This anticipated result is due to the high-binding affinity of PKS$_{5-24}$, which causes an

overall quenching of the conformational dynamics in the µs–ms time scale as observed with the PKI$_{5-24}$ inhibitor (Supplementary Fig. 16)[19]. What is unexpected is the presence of conformational dynamics in the PKA-C/ADP/pPKS$_{5-24}$ complex, which represents the ternary complex after phosphorylation (Supplementary Fig. 17). Conformational exchange is detected for several methyl groups located in both lobes. Remarkably, the DyCorr analysis shows that the conformational exchange dynamics is essentially uncorrelated (i.e., asynchronous). Finally, the exit complex (ADP-bound state) features an increase of these conformational exchanges involving several groups within the substrate binding site (Supplementary Fig. 18). Although several residues move with the same exchange rate, the motion is asynchronous for most of them and only show sparse correlations in the DyCorr map and the contrast with the ATPγN is particularly striking. To validate the impact of the synchronous motions of the enzyme, we analyzed the synchronicity of the slow motions of the side chain methyl groups for a dysfunctional mutant of PKA-C, where a single, allosteric mutation (Y204A) is able to reduce drastically the catalytic efficiency of the enzyme without changing the enzyme's structure[50]. The Y204 residue is located in the P + 1 loop and does not interact with either nucleotide or substrate. The mutation of the Tyr residue into Ala causes the disruption of a central electrostatic node involving Arg133 (D-helix), Glu230 (F-helix), Arg (P-2 site), and Glu170 (catalytic loop) that is critical for substrate recognition. Upon mutation, the enzyme's catalytic efficiency measured for the canonical Kemptide drops from 1.07 to 0.0025 µM$^{-1}$ s$^{-1}$. The low affinity for substrates prevents the measurement of dissociation constant and an accurate estimate of cooperativity. The analysis of methyl group RD for PKA-C$^{Y204A}$ shows that the enzyme becomes significantly more dynamic in the µs–ms time scale. However, the DyCorr map shows that the degree of inter-residue correlations (i.e., allosteric network) upon binding the nucleotide is rather different from that of the wild-

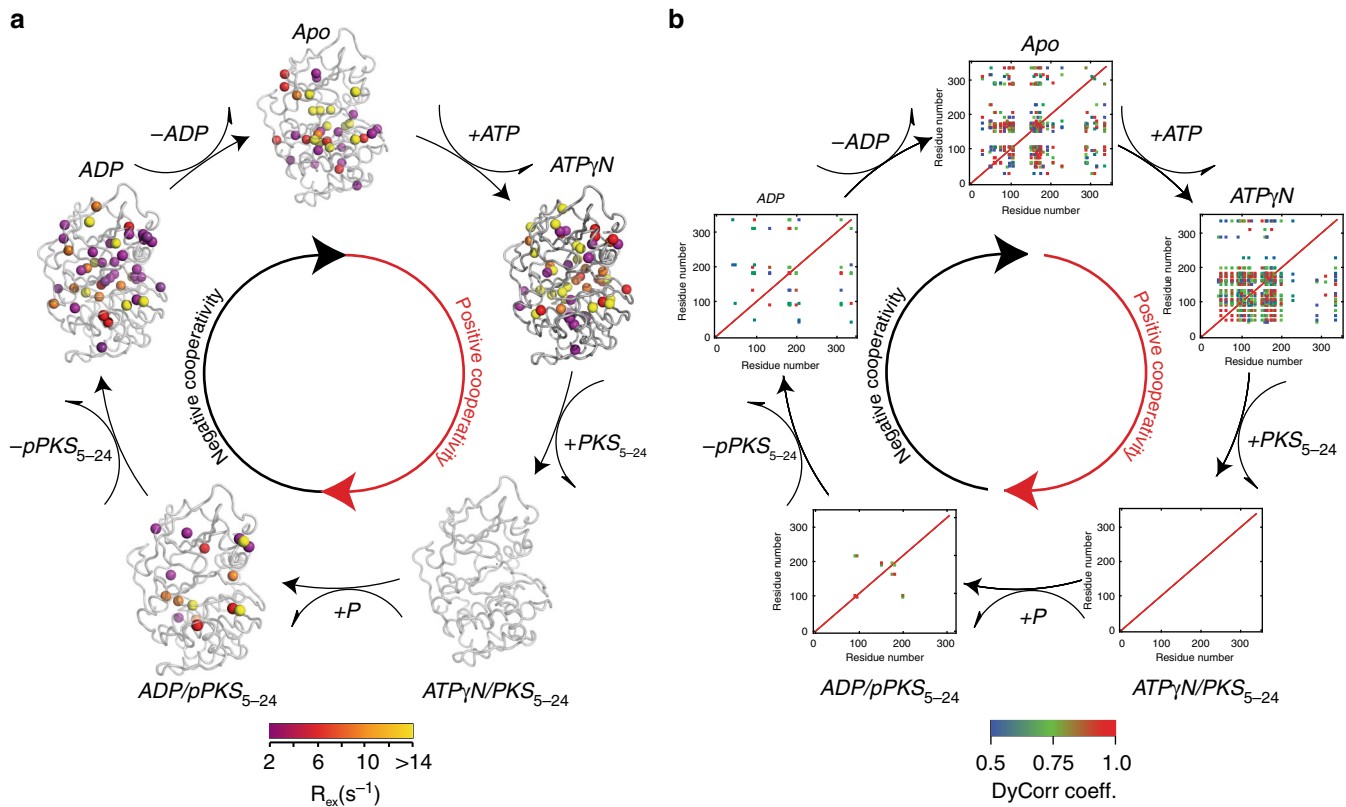

**Fig. 5** Synchronous and asynchronous motions of PKA-C along reaction. **a** Conformational exchange rates obtained using methyl CPMG experiments mapped onto the X-ray crystal structures: apo (PDB code: 4NTS), binary complex (ATPγN-bound, PDB code: 1BKX), ternary complex (ATP and PKS$_{5-24}$, PDB code: 4DG0), ternary/exit complex (ADP and pPKS bound, PDB code: 4IAF), and binary (ADP-bound, PDB code: 4NTT). **b** DyCorr maps of the different states of the kinase, showing the synchronization of motion upon nucleotide binding and the asynchronous dynamics in the ternary exit complex and ADP bound

type PKA-C (Supplementary Fig. 19). When compared with the DyCorr map in Supplementary Fig. 15, the nucleotide, which acts as an allosteric effector, is unable to establish the network of communication intra- and inter-lobes. Specifically, in the DyCorr maps of Supplementary Fig. 19 the correlations between Gly-rich loop, B- and C-helices and catalytic, DFG, and activation loops are absent, underscoring the loss of binding cooperativity[35]. Taken together, the conformational exchange data indicate that nucleotide binding to PKA-C synchronizes the motions, while asynchronous motions are present in the second part of the cycle destabilizing the intramolecular dynamic networks, and enabling product release.

## Discussion

Once unleashed from the regulatory subunit, PKA-C has most of its key structural elements assembled in an active state, with the activation loop phosphorylated at T197, the C-helix kept in place by the N- and C-tails and the formation of a stable salt bridge between K72 and E91. The binding of ATP completes the C-spine, bringing together the two lobes for catalysis. In this catalytically committed state, the ATP's adenine ring is sandwiched between Val57 and Leu173, with its γ-phosphate pointing toward the Ser acceptor (P site) of the substrate ready for phosphoryl transfer[15]. Indeed, the X-ray crystal structures offered high resolution pictures of the interactions occurring at the active sites, giving an enthalpic view of this crucial step. Yet, allosteric mutants of PKA-C, partially or completely inactive with impaired binding cooperativity, show well-folded

structures, almost superimposable to that of the wild-type enzyme[50]. To completely link structure to function, it is necessary to consider the intrinsic plasticity of kinases[44]. We previously proposed that ATP acts as an allosteric effector, shifting the kinase into a dynamically committed state[19]. What remained unexplored was the effects of ATP and ADP on the collective motions of the enzyme and how they affect positive and negative allosteric cooperativity along the enzymatic cycle. Elegant works by Wand and Kalodimos groups show that there is a direct correlation between the changes of overall entropy of binding measured thermodynamically and conformational entropy determined by NMR spectroscopy[10,11]. By fine-tuning protein's conformational entropy, it is possible to modulate ligand binding affinities. For PKA-C, we found that in the first part of the enzymatic cycle, the binding of ATP and substrate corresponds to an increase of conformational entropy that coincides with positive allosteric cooperativity with the substrate; while in the second part of the catalytic cycle a global rigidification of the enzyme's side chains (see $\Delta\Delta O^2$, Supplementary Fig. 12) corresponds to negative binding cooperativity. Unexpectedly, these variations of order parameters are highly correlated, suggesting that motions in the fast time scale are coupled throughout the enzyme. Stone and co-workers hypothesized that coupled motions would play a central role in binding cooperativity of enzymes[45]. They suggested that if the motions of groups of the two allosteric sites are coupled, the binding of one site would change the distribution of competent or non-competent states on the other site, thereby modulating positive and negative cooperativity, respectively[51]. Up to now,

these effects were not reported for a complete catalytic cycle. Our findings support this theory and highlight how conformational entropy drives enzymatic catalysis.

In addition to the conformational entropy, the kinases display significant breathing motions detected from the conformational exchanges in the μs–ms time scale. These motions exemplify the opening and closing of the active clefts for nucleotide and substrate binding[19,28], and abolishing them dramatically reduces the catalytic efficiency[35]. We found that these slow–time–scale conformational interconversions involve defined clusters of residues. As for the fast motions, the nucleotide orchestrates the conformational fluctuations throughout the entire enzyme, dynamically coupling the two binding sites in the small and large lobe. Specifically, the γ-phosphate and the coordinated $Mg^{2+}$ ion constitute electrostatic centers that stabilize the key salt bridges between K72 and E91, and position D184 for catalysis, thus engaging the β3 strand, the αC-helix, and the activation loop into a competent state for catalysis. This allosteric connection has been suggested by the recent community map analysis that reveals four communities converges around the ATP binding site (Supplementary Fig. 20)[34]. Upon releasing the γ-phosphate and Mg, the synchronous motions between these communities would be disrupted. Indeed, L74 at the β3 strand and L82 at αC-helix show distinct motions between the first and the second half of the cycle, underlying the agreement between NMR and the community map analysis[34]. These motions are present but they are not coupled when products are bound. A possible explanation is that slow and fast motions might act synergistically to facilitate product release. The $PKS_{5-24}$ peptide binding significantly attenuates the μs–ms motions, while enhancing the sub-nanosecond motions in the enzyme. $PKS_{5-24}$ has an unusually low value of $K_m$ reflecting its high affinity and a $k_{cat}$ of an order of magnitude lower with respect to native substrates[52]. Other peptide substrates display weaker affinity for the enzyme and the equilibrium does not shift to the fully closed state as compared to $PKS_{5-24}$[20]. Nonetheless, the enzyme retains significant conformational dynamics to transition between the open and closed states and facilitate product release[20].

Binding cooperativity has been recently shown for Src kinase as well[26]. In contrast to PKA-C, Src displays negative binding cooperativity with substrate and nucleotide; while the binding of products reveals positive binding cooperativity. While there is no experimental evidence supporting the role of coupled motions for Src, molecular dynamics simulations suggest the presence of allosteric networks of dynamic residues that mediate positive and negative cooperativity. It may be possible that altering the conformational dynamics and disrupting the allosteric coupling of these motions may reveal alternative avenues to manipulate kinase function and affect signaling.

## Methods

**Materials**. Adenosine 5′-triphosphate (ATP), γ-β-methyleneadenosine 5′-triphosphate (ATPγC), and Adenosine 5′-diphosphate (ADP) were purchased from Sigma Aldrich (St. Louis, MO, USA). Adenosine 5′-(β,γ-imido)triphosphate (ATPγN) was purchased from Roche Diagnostics (Indianapolis, IN, USA)

**Sample preparation**. Recombinant catalytic subunit of PKA was expressed in BL21 DE3 cells[18] at 24 °C and recombinant His-Riiα(R213K) was expressed in LB at 24 °C[53]. Purification of PKA-C was performed using the His₆-RIIα(R213K) subunit and a second purification step was performed by using a HiTrap SP cation exchange column[54]. Peptides were synthesized and cleaved using standard Fmoc chemistry on a CEM microwave peptide synthesizer and the crude peptide was purified by reversed-phase HPLC using a Supelco C18 semipreparative column[20]. Phosphorylated $PKS_{5-24}$ was produced by dissolving 400 μM of $PKS_{5-24}$ in a reaction mixture containing 50 mM MOPS, 80 mM $MgCl_2$, 15 mM ATP, and 0.5 μM of PKA-C at pH 7.0. The mixture was shaken in room temperature for 1 ½ hours and repurified using a Supelco C18 semipreparative column. Peptide

concentrations and identity was determined by elemental amino acid analysis (Texas A&M University, College Park, TX).

**ITC measurements**. Active enzyme was collected and exchanged into 20 mM MOPS, 90 mM KCl, 10 mM DTT, 10 mM $MgCl_2$, 1 mM $NaN_3$ at pH 6.5. Final concentrations for ITC measurements were 11.4–15 μM of PKA-C for titrations with peptide and 124 mM of PKA-C for titrations with ADP. PKA-C was quantified by $A_{280} = 52060\ M^{-1}\ cm^{-1}$. Corresponding nucleotide (ADP, ATPγC, and ATPγN) was added to a final concentration of 2 mM for the nucleotide bound state. ITC measurements were performed with a Microcal VP-ITC (Northampton, MA, USA) or a TA NanoITC (New Castle, DE, USA) instrument at 300 K. For the VP-ITC, 2 mL of PKA-C was used for each experiment and 280 μL of 140–313 μM of $PKS_{5-24}$ and 140–392 μM $pPKS_{5-24}$ in the titrant syringe. For NanoITC, 300 μL of PKA-C was used for each experiment and 50 μL of 2.4 mM of ADP in the titrant syringe. All experiments were performed in triplicate. The heat of dilution of the ligand to the buffer was taken into account by measuring the heat of dilution of the ligand to the buffer. Binding was assumed to be 1:1 and was analyzed using the Wiseman Isotherm[9] using the NanoAnalyze (TA instruments, New Castle, DE) software.

$$\frac{d[MX]}{d[X_{tot}]} = \Delta H^\circ V_0 \left[ \frac{1}{2} + \frac{1 - (1 + r)/2 - R_m/2}{\left(R_m^2 - 2R_m(1 - r) + (1 + r)^2\right)^{1/2}} \right], \qquad (1)$$

where the change of the total complex, $d[MX]$ with respect to the change of the ligand concentration, $d[X_{tot}]$ is dependent on $r$, the ratio of the $K_d$ with respect to the total protein concentration, and $R_m$, the ratio between the total ligand and total protein concentration. The free energy of binding was determined using the following:

$$\Delta G = RT \ln K_d, \qquad (2)$$

where $R$ is the universal gas constant and $T$ is the temperature at measurement (300 K). Calculations for the cooperativity constant ($\sigma$) were calculated as follows:

$$\sigma = \frac{K_{d,Apo}}{K_{d,nucleotide}}, \qquad (3)$$

where $K_{d,Apo}$ is the $K_d$ of $PKS_{5-24}/pPKS_{5-24}$ to the apo enzyme and $K_{d,nucleotide}$ is the $K_d$ of $PKS_{5-24}/pPKS_{5-24}$ to the nucleotide bound enzyme.

**NMR measurements**. Samples for $^2H$, $^{13}C$ IVL $^{15}N$ labeled PKA-C was expressed in E. coli bacteria and purified using the His₆-RIIα(R213K) subunit and a second purification step was performed by using a HiTrap SP cation exchange column[18,54]. Assignments for the methyl groups of the IVL residues of PKA-C have been previously reported[54]. Effective final sample concentrations for chemical shift mapping were 0.2–0.25 mM in 20 mM $KH_2PO_4$, 90 mM KCl, 10 mM DTT, 10 mM $MgCl_2$, and 1 mM $NaN_3$ at pH 6.5. Titrations with $pPKS_{5-24}$ started with 12 mM of nucleotide (ADP or ATPγN). Addition of $PKS_{5-24}$ was used for a minimum of twofold molar excess of peptide and titration with ADP was carried out to a final concentration of 12 mM. NMR assignments on the apo, nucleotide bound (ATPγN) and ternary (ATPγN and $PKI_{5-24}$) were carried out on a 850 MHz Bruker Advance III spectrometer and described elsewhere[19]. All $^1H$-$^{13}C$ HMQC experiments were carried out on an 850 MHz Bruker Advance III spectrometer with a TCI cold probe operating at 300 K. The magnitude of the chemical shift perturbation was calculated as follows:

$$\Delta\delta_{HX} = \sqrt{\Delta\delta_H^2 + c * \Delta\delta_X^2}, \qquad (4)$$

where $X$ is the heteronuclei indirectly measured and $c$ is a scaling factor (0.25 for $X = {}^{13}C$ and 0.154 for $X = {}^{15}N$).

**Spin-relaxation experiments on methyl order parameter**. Measurement of the order parameter was performed using triple quantum based $^1H$ relaxation violated coherence transfer cross-correlation experiment[41,42]. The $^{13}CH_3$ IVL methyl-labeled $^2H$ PKA-C sample was concentrated to 230 μM with 12 mM ADP and 1.25 mM $pPKS_{5-24}$ for the product complex, 230 μM with 12 mM ATPγN and 2.0 mM $PKS_{5-24}$ and 250 μM with 12 mM ADP. All the samples were checked by DLS to confirm the sample conditions and determine the global rotational correlation time ($\tau_c$) and set to a value of 25 (ternary form) or 29 ns (ADP bound state). Single and triple quantum filtered spectra were acquired with relaxation delays ($T$) of 3, 6, 9, 12, 15, 20, 25, 30, and 35 ms using 2048 × 160 complex data points using a recycle delay of 2 s. A nonlinear fit to the ratio of the intensities was used to extract the cross-correlation rate ($\eta$):

$$\frac{I_a}{I_b} = \frac{3}{4} \frac{\eta \tanh\left(\sqrt{\eta^2 + \delta^2}\,T\right)}{\sqrt{\eta^2 + \delta^2} - \delta \tanh\left(\sqrt{\eta^2 + \delta^2}\,T\right)}, \qquad (5)$$

where $I_a$ is the intensity for the triple quantum filtered spectra, $I_b$ is the intensity for the single-quantum filtered spectra, and $\delta$ is a constant accounting for the coupling of the methyl group with external protons. Error analysis was performed using Monte-Carlo sampling. From the cross-correlation rate the order parameter was estimated using the following:

$$\eta \approx \frac{9}{10}\left(\frac{\mu_0}{4\pi}\right)^2 [P_2(\cos\theta_{HH})]^2 \frac{S^2\gamma_H^4\hbar^2\tau_c}{r_{HH}^6}, \tag{6}$$

where $\mu_0$ is the permittivity of vacuum, $P_2$ is the second order Legendre polynomial, $\theta_{HH}$ is the angle between the vector connecting two intra-methyl protons and the methyl rotational axis, and $r_{HH}$ is the distance between two protons within the methyl group. All the methyl order parameter values are listed in the Supplementary Data 1.

**Methyl CPMG RD experiments.** Single-quantum $^{13}C$ CPMG experiments were carried out on Bruker Avance 700 MHz and Avance III 850 MHz spectrometers. Spectra were collected in an interleaved fashion with CPMG field strengths of 50, 100, 150, 200, 250, 300, 400, 500, 600, 800, and 1000 Hz with replicate experiments performed on 50, 200, and 1000 Hz with a constant time delay of 40 ms. All data were processed using NMRpipe and peak intensities were picked using Sparky. The peak intensities were converted to transverse decay rates, $R_{2,eff}$[55]. GUARDD[49] was used to fit the relaxation data to the Carver–Richards equation[56], which describes the dependence of the relaxation contribution of chemical exchange to transverse relaxation, $R_{ex}$, to the exchange rate and the population of the second state for a two-state exchange process.

$$R_2\left(\frac{1}{\tau_c}\right) = R_2^0 + \frac{1}{2}\left(k_{ex} - \frac{1}{\tau_c}\cosh^{-1}\left[D_+\cosh(\eta_+) - D_-\cos h(\eta_-)\right]\right), \tag{7}$$

$$D_\pm = \frac{1}{2}\left(\pm 1 + \frac{\psi + 2\Delta\omega^2}{(\psi^2 + \xi^2)^{1/2}}\right), \tag{8}$$

$$\eta_\pm = \frac{\tau_{cp}}{\sqrt{2}}\left((\psi^2 + \xi^2)^{1/2}\right)^{1/2}, \tag{9}$$

$$\psi = k_{ex}^2 - \Delta\omega^2, \quad \xi = -2\Delta\omega(p_A - p_B). \tag{10}$$

Where $p_A$ and $p_B$ are the population of the two states, $\Delta\omega$ the chemical shift difference of the nuclei between the two states, $\tau_{cp}$ is the time between the $\pi$ pulses, $k_{ex}$ is the sum of on and off exchange rates and $R_2^0$ is the intrinsic transverse relaxation rate. CPMG fit was performed by minimizing the function, $\chi^2$[49].

**Analysis of the chemical shift perturbations.** We employed the COordiNate ChemIcal Shift bEhavior (CONCISE)[27] method to monitor trajectories of chemical shifts and measure the change in equilibrium position associated with each PKA-C state (ADP, ATPγN ADP/pPKS$_{5–24}$, and ATPγN/PKS$_{5–24}$). A basis set comprising of the apo, ATPγN, ATPγN/PLN$_{1–19}$, and ATPγN/PKI$_{5–24}$ states were used to define the open to closed states of the kinase[33]. CONCISE was applied to side chain IVL methyl groups from $^1H$–$^{13}C$ methyl-TROSY experiments. Principal component analysis (PCA) was used to identify a set of residues whose chemical shifts respond linearly to the conformational transition. Each one of these residues provides a measure of the equilibrium position for every PKA-C construct in form of scores along the first principal component (PC1), while the spread around the linear trajectory is given by the second principal component (PC2). The equilibrium position along the open to closed trajectory for a given construct is given by the average of the PC1-scores over all linear residues. To identify the residues whose chemical shifts follow a linear pattern, a threshold of 3.0 for the ratio of the standard deviations of PC1 over PC2 was used and residues that were affected by chemical shifts perturbations below 0.05 ppm were also discarded (see ref. 27 for details on the threshold calibration). After these thresholds were applied, a total of 48–49 side chain methyl resonances formed the subset that was used to trace the equilibrium position of each state (Supplementary Table 3) using the apo, ATPγN, ATPγN/PLN$_{1–19}$, and ATPγN/PKI$_{5–24}$ states as basis sets. To identify the largest group of residues that respond to ligand binding in a correlated fashion, we have used chemical shift covariance analysis[31] (CHESCA) with a filtering condition on chemical shift perturbation level. We have selected the residues with maximum vector distance of various states greater than a cutoff. The maximum vector distance is evaluated using,

$$\text{Maximum vector distance} = \text{Max}\left(\sqrt{(0.25\delta_{Cij})^2 + (\delta_{Hij})^2}\right)_{i,j=1,2,3...n},$$

where $\delta_{Cij}$ and $\delta_{Hij}$ are the difference in $^{13}C$ and $^1H$ chemical shift of $i$th and $j$th state and $n$ is the numbers of states.

The CHESCA correlation matrix with a distance cutoff of 0.1 and 0.05 are shown in Supplementary Fig. 8 and used to build a dendrogram through

hierarchical clustering (Supplementary Fig. 9). Residues that were highly correlated (with a $r_{ij} > 0.9$) were identified and the correlations were mapped onto the structure of PKA-C (PDB: 1ATP) using a custom built PyMol script generator in Matlab. The CHESCA data are available in the Supplementary Data 2.

**Dynamical correlation (DyCorr) map.** In order to study the μs–ms time scale domain motion in protein one can use the group fitting of dispersion profiles of which are closer in proximity or allosterically connected. The individual dispersion curves in a group are fitted for a common exchange rate and population. In the standard procedure of group fitting, structural information is used to group proximal residues. This may rule out the allosterically connected dynamic residues also the reduced degree of freedom force individual residues to fit to a common dynamic parameter. Here, we propose a novel scheme for analyzing chemical exchange dynamics. The method maps the residues into a space defined by dynamic parameters and does not require the global fitting of residues. To generate the DyCorr maps, we apply the following protocol:

1. All the CPMG dispersion profiles are fitted using Carver–Richard equation and generate chemical exchange parameters ($k_{ex}$, $pE$, $\Delta\omega$, $R_2^0$) for all the residues. ($k_{ex}$ : chemical exchange rate, $pE$ : population of excited state, $\Delta\omega$ : chemical shift difference between ground and excited state and $R_2^0$ : intrinsic transverse relaxation rate).
2. The parameters ($k_{ex}$, $pE$) carry μs–ms dynamics information, and are used for generating DyCorr maps. The chemical shift difference ($\Delta\omega$) and intrinsic transverse relaxation rate ($R_2^0$) are completely independent of motion in this scale.
3. The on and off exchange rates ($k_{on}$, $k_{off}$) are generated from ($k_{ex}$, $pE$) using the relations

$$k_{on} = pE \times k_{ex}, \tag{11}$$

$$k_{off} = pG \times k_{ex}. \tag{12}$$

4. All the residue with ($k_{on}$, $k_{off}$) are mapped in to $k_{on}$ – $k_{off}$ space, where every dynamic residue is represented by a point.
5. Find the distance of each individual points from all the others. The relative distances for all the pair of points are evaluated by normalizing by the distance to the middle point of the vector from the origin as shown in the Supplementary Fig. 21, where $k_{on}$ – $k_{off}$ are mapped for the kinase. Here each point in the graph represents a CPMG dispersion curve. The relative distance ($\kappa_{ij}$) is evaluated for all the pairs.
6. From the relative distance, evaluate the relative proximity ($\eta_{ij}$) using a linear or nonlinear function. The relative proximity is a measure of how close two residues are in $k_{on}$ – $k_{off}$ space, where $\eta_{ij} = 1$ represents the closest residues and $\eta_{ij} = 0$ represents the farthest ones. Here we have used the following nonlinear function (Eq. (13)) to evaluate $\eta_{ij}$.

$$\eta_{ij} = \frac{(\kappa_m - \kappa_{ij})}{\kappa_m(\kappa_{ij} + 1)}, \tag{13}$$

where $\kappa_m$ is the maximum value of $\kappa_{ij}$. Note that the maximum values for relative distance ($\kappa_m$) is 2 (which is length of diagonal divided by the distance to middle point from origin). The above equation (Eq. (13)) is a simple nonlinear function of $n_{ij}$ varying from 0 to 1, for $k_{ij}$ in the range [0, 2]. One can use a simple linear function (Eq. (14)) or cosine function for the same.

$$\eta_{ij} = 1 - 0.5\kappa_{ij}. \tag{14}$$

DyCorr map is the map of relative proximity ($\eta_{ij}$) of residues where a maximum in $ij$th element represent residue $i$ and $j$ are close in μs–ms dynamics.

The dynamic correlation values of the DyCorr maps are available in Supplementary Data 3.

**Dynamic light scattering.** Samples were incubated at 298 K and a Malvern Zetasizer μV was used to record in triplicate the autocorrelation function of the scattered light. The Z-average diameter, reporting on the hydrodynamic radius of the protein, was from an average of the measured intensity autocorrelation function using the Zetasizer software Version 6.34. Assuming isotropic molecular tumbling, the rotational correlation time ($\tau_c$) was calculated using the Stokes–Einstein equation at 300 K.

**System setup and MD simulations.** The ternary complexes were prepared from the crystal structure of PKA in a closed ternary conformation bound with PKI$_{5–24}$

inhibitor peptide, ATP, and two $Mn^{2+}$ ions (PDB code 3FJQ). The $Mn^{2+}$ ions were changed to $Mg^{2+}$ and the inhibitory peptide was mutated at the P and P-1 sites into the pseudo-substrate peptide $PKS_{5-24}$. The model was further processed in Maestro (Schrodinger) where the Protein Preparation Wizard was used to add counter ions and model histidine, yielding HIP for H87, HIE for H142, and HID for the remain histidines, and the C199 was modeled as negatively charged[34]. For the ternary/exit complex a phosphate group was added at the P site and ADP was derived from the ATP model. Hydrogens were added and the models were solvated in a cubic box of TIP4P-EW water[57] with a 10 Å buffer in AMBERtools[58]. Parameters from the AMBER Parameter Database[59] were used for ATP[60], ADP[60], phosphothreonine[61], and phosphoserine[61].

AMBER16[58] was used for energy minimization, heating, and equilibration steps, using the central processing unit code for minimization and heating and graphics processing unit (GPU) code for equilibration. Systems were minimized by 500 steps of hydrogen-only minimization, 500 steps of solvent minimization, 500 steps of sidechain minimization, and 5000 steps of all-atom minimization. Systems were heated from 0 to 300 K linearly over 500 ps with 2 fs time-steps and 10.0 kcal mol Å position restraints on protein and ligands. Temperature was maintained by the Langevin thermostat. Constant pressure equilibration with a 10 Å nonbonded cutoff was performed with 100 ps of protein and ligand restraints followed by 100 ps without restraints. An 8 Å cut-off was used for short-range nonbonded interactions and particle mesh Ewald[62] was used to treat long-range electrostatic interactions during a final 50 ns of simulation. Production simulations were performed on GPU enabled AMBER16 as above in triplicate for a total aggregate simulation time of 5.4 μs for each complex under identical conditions for comparative purposes. The first 100 ns of each simulation were removed prior to analysis.

**Conformational entropy calculations**. To compare conformational entropy for each residue, MD trajectories of the ternary complexes and previously published apo and binary states[34] were used at 120 ps intervals. Trajectories were aligned by C-lobe residues 128–300. Cartesian coordinates of backbone Cα and representative side-chain atoms of each residue were used for the entropy estimations[34]. Self-information or entropies were estimated using modified Mutinf scripts[34] by a histogram-based entropy expansion method for each residue, where the first-order entropy approximation was corrected for pairwise mutual information for each degree of freedom. Briefly, for each residue's Cα and representative side-chain atom one- and two-dimensional uniformly binned histograms were used to determine the positional self- and joint-entropies. Corrections to the mutual information due to undersampling and nonzero mutual information in randomly generated independent data were made. Final entropies were calculated by removing contributions from correlated motions (mutual information) from the entropies (first-order terms).

**Reporting summary**. Further information on experimental design is available in the Nature Research Reporting Summary linked to this article.

## Data availability

All data illustrated in this study are included in the Supporting Information Files and codes are available from the authors upon reasonable request. The source data underlying Fig. 1d and Supplementary Fig 10 are provided as a Source Data file.

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

## Acknowledgements

This work is supported by the NIH (GM100310 to G.V., T32AR007612 to J.K., and T32CA009523 to P.A.). NMR Experiments were carried out at the Minnesota NMR Center. We thank Prof. N. Levinson, and Prof. A. Cembran for helpful discussions. We thank Prof. Robert Geraghty for the access to the VP-ITC equipment.

## Author contributions

G.V. and S.S.T. conceived and directed all the research; Y.W. and M.V. analyzed the experimental data; J.K. collected NMR and ITC data, analyzed NMR and ITC data, and helped with the initial draft of the manuscript. G.L. collected NMR data; L.A. and P.A. conducted MD simulations and analyzed the simulation data. G.V., Y.W., and M.V. prepared the final draft of the manuscript with inputs and feedbacks from all authors.

## Additional information

**Competing interests:** The authors declare no competing interests.

**Journal Peer Review Information**: *Nature Communications* thanks the anonymous reviewer(s) for their contribution to the peer review of this work. Peer reviewer reports are available

