## [Peer Review File · Nature Communications]

Reviewers' comments:

Reviewer #1 (Remarks to the Author):

The authors present a very thorough analysis of the protein dynamics of PKA kinase at the different ligation states relevant to catalysis. The study is an important extension of previous work that qualitatively and quantitatively yields much needed new insights into the mechanism of this well studied enzyme. The experiments are performed and analyzed thoroughly and comprehensively describe the findings.

I recommend the study for publication. A few minor points might be considered by the authors to make some of the very advanced methods more palatable to a broader readership.

- the authors perform calorimetric studies to determine the relative stability of liganded states relative to apo. Since they obtain also entropic data that could be correlated with the NMR and MD data, it might be helpful to have a side-by-side comparison.
- the authors compare wt to an allosteric mutant by NMR dynamics and find marked differences in the correlated dynamic networks. The discussion of the effect of the mutation could be more detailed and available information on catalysis and ITC could be included as well (if available).

Reviewer #2 (Remarks to the Author):

Wang et al. reported findings for the cooperative substrate binding in the kinase PKA-C, using a combined approach of thermocalorimetry, NMR, and molecular dynamics. They found that conformational entropy characterized with subnanosecond dynamics, along with synchronous and asynchronous motions of the enzyme happening in the micro- to millisecond timescale, are factors determining the positive and negative allosteric cooperativity during reactants binding and products release, respectively. The manuscript is well written. The results are very comprehensive, complete, and potentially will inspire others' work in the field.

However, the authors are a bit overselling their findings. Although conformational entropy might influence substrate and nucleotide binding in PKA-C, from what the authors have shown it is not evident that the entropy "determines" the divergent cooperativities. Figure 2 shows that changes of conformational entropy are in the same direction upon binding both ATP and ADP, although the magnitude of change in the former is larger. This indicates that binding either ATP or ADP "prepays" the entropic cost needed to bind subsequent substrates, not explaining why binding ADP makes the binding of the phosphorylated product more difficult. Actually, the results suggest that the deterministic factor may be the motions observed in a larger timescale: Figure 5B clearly shows that changes of dynamical correlations captured in the micro- to millisecond dynamics from Apo to ATP(γ)N and ADP have opposite trends, consistent with the different cooperativities.

"Prepaying" entropic cost for positive allosteric cooperativity (or opposite) is not a new concept. The theoretical work was completed over 30 years ago and the model has been verified by both experimental and theoretical studies. The authors should make this point clear with citing proper references. For example,

Cooper & Dryden, *Europ. Biophys. J.* (1984) 11: 103-109.
Popovych et al., *Nat. Struct. Mol. Biol* (2006) 13: 831-838.
Petit et al., *PNAS* (2009) 106: 18249-18254.
Law et al., *PNAS* (2014) 111: 12067-12072.

Barman & Hamelberg, J. Phys. Chem. B (2016) 120:8405-8415.

Reviewer #3 (Remarks to the Author):

The paper by Veglia and co-workers addresses the importance of conformational entropy changes in the regulation of enzyme function.

By focusing on dynamics, as a sign of disorder, rather than on structure, entropy changes, and their relative importance in binding and conformational transitions, can be assessed.

The authors use laborious NMR measurements, Isothermal Titration Calorimetry and molecular dynamics simulations to characterize the entire enzymatic cycle of the catalytic subunit of cAMP-dependent PKA-C. The combined use of ITC and NMR is emerging as a tool to estimate conformational contributions to the free energy of binding.

Compared to pioneering works in this field which are cited by the authors, the chemical shifts analysis which leads to the population analysis is new and provides further detail on the conformational equilibria.

This is an important contribution, the paper is well-written and the many informations reported are well organized for the reader. The entire enzymatic cycle is addressed which adds reliability to the results. The overall methodology could be a reference to other authors for similar studies.

I found all the procedures and the discussion convincing and thus I have no particular comments on the experimental part. I have only a major comment on the simulations.

Reading the paper, one gets the impression that molecular dynamics simulations are underused compared to their potential. Also, the exact analysis performed is not clearly described.

A cut-off of 8 Å is used, which is very short, I imagine to speed up the simulation. This requires however a word of caution. If PME has been used in conjunction with the short cutoff, then this must be mentioned.

Assuming that the choice of parameters did not lead to major inaccuracies, which would however have not gone unnoticed by the authors, the exact way entropy is computed is not clear.

The authors report 5.4 μs aggregated simulation time for each of the simulated complexes, which implies extensive conformational sampling.

The authors say that "cartesian self entropies of backbone and representative side-chain atoms of each residue were determined by a corrected histogram entropy estimate". It is not clear exactly what has been done, or which program has been used. The reference of Kilian et al., is the reference for the Mutual information expansion of Gilson, whereas the theoretical reference by McClendon et al., refer to the Jacobson approach. Since the bond (and to a great extent also angle) contributions to entropy change little with conformation (as shown by Karplus and others), the best way to compute conformational entropy changes is by considering torsional angles (as done in both references). The term "cartesian self-entropy" makes one think of an approximate way to estimate entropy based on the mutual information (as in the Jacobson and Gilson approaches) between distributions in cartesian space (i.e. based on position correlations). This is further suggested by the superposition performed before the analysis.

In any case a precise explanation of what has been done is due. If there are space limitations this

could be placed in the supplementary materials.

Perhaps for future application it is worth to consider that the estimation of conformational entropy from molecular dynamics snapshots has made great progress in the last years, thanks to the nearest neighbour method of Gilson and co-workers and the maximum information spanning tree method of Tidor and co-workers, and programs are available (e.g. pdb2entropy or PARENT) that can provide accurate residue-by-residue entropy estimates.

Minor issues

There are some typos. Also, I believe, thermodynamics is singular, so "While thermodynamics allow ..." should read "While thermodynamics allows... "

line 318 - exchange ->exchanges

line 348 - work -> ...works

line 501 - CONCISE was to side chain... -> CONCISE was applied to side chain..

RESPONSE TO REVIEWERS

Reviewer #1

AU: We thank this reviewer for her/his positive and constructive comments. In this revised version, we implemented the changes suggested and improved our manuscript based on this critique.

REV#1: [...] Since they obtain also entropic data that could be correlated with the NMR and MD data, it might be helpful to have a side-by-side comparison.

AU: We now include a new table (Supplementary Table 2) in the Supplemental Information with side-by-side comparison of the entropy changes between these techniques. Note also that the changes in the entropy of the enzyme were calculated using two different models. We included a brief description for these calculations in the Supplementary Information.

REV#1 - the authors compare wt to an allosteric mutant by NMR dynamics and find marked differences in the correlated dynamic networks. The discussion of the effect of the mutation could be more detailed and available information on catalysis and ITC could be included as well (if available).

AU: We added a few paragraphs on page 12, describing the Dycorr map as well as kinetic and thermodynamic data for the Y204A mutant available from previous publications ^{1,2}.

Reviewer #2:

AU: We would like to thank this reviewer for the positive comments and bring to our attention the seminal papers on the pre-paid entropy cost, which we now include in the citations. Below is our response to her/his important critiques.

REV#2: However, the authors are a bit overselling their findings. Although conformational entropy might influence substrate and nucleotide binding in PKA-C, from what the authors have shown it is not evident that the entropy “determines” the divergent cooperativities.

AU: Following this suggestion, we have toned it down the text.

REV#2: Figure 2 shows that changes of conformational entropy are in the same direction upon binding both ATP and ADP, although the magnitude of change in the former is larger. This indicates that binding either ATP or ADP “prepays” the entropic cost needed to bind subsequent substrates, not explaining why binding ADP makes the binding of the phosphorylated product

more difficult. Actually, the results suggest that the deterministic factor may be the motions observed in a larger timescale: Figure 5B clearly shows that changes of dynamical correlations captured in the micro- to millisecond dynamics from Apo to ATP(γ)N and ADP have opposite trends, consistent with the different cooperativities. “Prepaying” entropic cost for positive allosteric cooperativity (or opposite) is not a new concept.

AU: We thank this reviewer for this comments. We have addressed her/his comments in the revised version of the manuscript. Briefly, we revised Supplementary Figure 12 and its legend to clarify that the changes in the conformational entropy (i.e., changes in the methyl group order parameters) support our original conclusions. While both the variation of the order parameters for the formation of the exit complex (PKA-C/ADP/pPKS) and the Michaelis-Menton complex (PKA-C/ATP/PKS) follow the same trend (i.e., on average the changes have both negative signs); the difference of ΔO^2 between the two binding steps ($\Delta\Delta O^2$) is positive, indicating that the exit complex is more rigid than the Michaelis-Menton complex. Note that the $\Delta\Delta O^2$ values are significantly higher than those previously found by Kalodimos and co-workers for CAP, indicating a larger contribution of entropy to facilitate product release. Indeed, we agree with the view of this reviewer about the pre-paid entropy cost, which occurs upon the formation of the PKA-C/ATP and PKA-C/ADP complexes. We included a few paragraphs in the Discussion.

REV#2: The theoretical work was completed over 30 years ago and the model has been verified by both experimental and theoretical studies. The authors should make this point clear with citing proper references.

AU: We thank this reviewer for pointing out these previous works. Accordingly, we modified our text, citing these important papers that she/he suggested

Reviewer #3

AU: We thank this reviewer for careful reading of our manuscript. Below are the point-by-point responses to her/his concerns.

REV#3: A cut-off of 8 Å is used, which is very short, I imagine to speed up the simulation. This requires however a word of caution. If PME has been used in conjunction with the short cutoff, then this must be mentioned. Assuming that the choice of parameters did not lead to major inaccuracies, which would however have not gone unnoticed by the authors, the exact way entropy is computed is not clear.

AU: We thank this reviewers for pointing out this. Indeed, the 8 Å cut-off is only a hard cut-off for the short-range van der Waals' interactions. For electrostatics, we considered both short-range and long-range terms. The electrostatics interactions beyond 8 Å are handled by the Particle Mesh Ewald Method in AMBER. We now clarify this point in the Method section.

REV#3: The authors report 5.4us aggregated simulation time for each of the simulated complexes, which implies extensive conformational sampling. The authors say that "cartesian self entropies of backbone and representative side-chain atoms of each residue were determined by a corrected histogram entropy estimate". It is not clear exactly what has been done, or which program has been used. The reference of Kilian et al., is the reference for the Mutual information expansion of Gilson, whereas the theoretical reference by McClendon et al., refer to the Jacobson approach. Since the bond (and to a great extent also angle) contributions to entropy change little with conformation (as shown by Karplus and others), the best way to compute conformational entropy changes is by considering torsional angles (as done in both references). The term "cartesian self-entropy" makes one think of an approximate way to estimate entropy based on the mutual information (as in the Jacobson and Gilson approaches) between distributions in cartesian space (i.e. based on position correlations). This is further suggested by the superposition performed before the analysis. In any case a precise explanation of what has been done is due. If there are space limitations this could be placed in the supplementary materials.

AU: Based on this reviewer's comments, we significantly reworked the Method section and revised the citations. Specifically, our method to determine entropy was adapted from McClendon *et al.* 2014 ³, which are modified versions of the previous version [see McClendon 2009 ⁴]. Based on this previous work, Cartesian coordinates are able to capture correlated motions in semirigid regions and enable a direct comparison with published PKA-C simulations carried out for both apo and binary complex (McClendon 2014). Note also that the diagonal elements of the mutual information matrix plus positional self-information correction yield the same corrected entropy term as described in Killian 2007 ⁵. Since we analyzed the relative changes in entropy between nearly identical systems, errors in the entropy calculations due to higher-order/long-range correlations are negligible.

REV#3: Perhaps for future application it is worth to consider that the estimation of conformational entropy from molecular dynamics snapshots has made great progress in the last years, thanks to the nearest neighbour method of Gilson and co-workers and the maximum information spanning tree method of Tidor and co-workers, and programs are available (e.g. pdb2entropy or PARENT) that can provide accurate residue-by-residue entropy estimates.

AU: This is a good suggestion and will be part of our future endeavors.

Minor issues

There are some typos. Also, I believe, thermodynamics is singular, so "While thermodynamics allow ..." should read "While thermodynamics allows... "

line 318 - exchange ->exchanges

line 348 - work -> ...works

line 501 - CONCISE was to side chain... -> CONCISE was applied to side chain..

AU: These typos have been corrected in the revised manuscript.

1. Srivastava, Atul K. et al. Synchronous Opening and Closing Motions Are Essential for cAMP-Dependent Protein Kinase A Signaling. *Structure* **22**, 1735-1743 (2014).
2. Ahuja, L.G., Kornev, A.P., McClendon, C.L., Veglia, G. & Taylor, S.S. Mutation of a kinase allosteric node uncouples dynamics linked to phosphotransfer. *Proc Natl Acad Sci U S A* **114**, E931-E940 (2017).
3. McClendon, C.L., Kornev, A.P., Gilson, M.K. & Taylor, S.S. Dynamic architecture of a protein kinase. *Proc Natl Acad Sci U S A* **111**, E4623-31 (2014).
4. McClendon, C.L., Friedland, G., Mobley, D.L., Amirkhani, H. & Jacobson, M.P. Quantifying Correlations Between Allosteric Sites in Thermodynamic Ensembles. *Journal of Chemical Theory and Computation* **5**, 2486-2502 (2009).
5. Killian, B.J., Kravitz, J.Y. & Gilson, M.K. Extraction of configurational entropy from molecular simulations via an expansion approximation. *The Journal of Chemical Physics* **127**, 024107 (2007).

REVIEWERS' COMMENTS:

Reviewer #2 (Remarks to the Author):

The authors have addressed part of my concerns but the interpretation of order parameter changes (Supplementary Figure 12 and Figure 2) is still a bit confusing. For example, in page 9, line 195, it says: "the difference in the average value of the order parameters between ADP bound and ADP/pPKS bound does not follow the same trend as ATP[gamma]N bound to the ATP[gamma]N/pPKS bound." But, in the later part of the same paragraph, it says: "While the changes in the order parameters follow the same sign for both the Michaelis Menton and product formation complex, ..." This inconsistency should be fixed. Also, I would recommend avoid saying "signifying a negative change in entropy and reduced binding affinity" (p9, line 211), because $\Delta\Delta O_2$ indicates different magnitudes of entropic changes not necessarily the opposite directions of the changes.

REVIEWERS' COMMENTS:

Reviewer #2 (Remarks to the Author):

The authors have addressed part of my concerns but the interpretation of order parameter changes (Supplementary Figure 12 and Figure 2) is still a bit confusing. For example, in page 9, line 195, it says: “the difference in the average value of the order parameters between ADP bound and ADP/pPKS bound does not follow the same trend as ATP[gamma]N bound to the ATP[gamma]N/pPKS bound.” But, in the later part of the same paragraph, it says: “While the changes in the order parameters follow the same sign for both the Michaelis Menton and product formation complex, ...” This inconsistency should be fixed. Also, I would recommend avoid saying “signifying a negative change in entropy and reduced binding affinity” (p9, line 211), because $\Delta\Delta O_2$ indicates different magnitudes of entropic changes not necessarily the opposite directions of the changes.

RE: We appreciate this reviewer for careful reading of this revised manuscript and for pointing out this inconsistency in the text, which we revised accordingly.